# Challenges in Liver Transplantation for Hepatocellular Carcinoma: A Review of Current Controversies

**DOI:** 10.3390/cancers16173059

**Published:** 2024-09-02

**Authors:** Ezequiel Mauro, Marco Sanduzzi-Zamparelli, Gabrielle Jutras, Raquel Garcia, Alexandre Soler Perromat, Neus Llarch, Victor Holguin Arce, Pablo Ruiz, Jordi Rimola, Eva Lopez, Joana Ferrer-Fàbrega, Ángeles García-Criado, Jordi Colmenero, Jennifer C. Lai, Alejandro Forner

**Affiliations:** 1Liver Oncology Unit, Liver Unit, ICMDM, Hospital Clinic Barcelona, 08036 Barcelona, Spain; msanduzzi@clinic.cat (M.S.-Z.); nllarch@clinic.cat (N.L.); 2Barcelona Clinic Liver Cancer (BCLC) Group, IDIBAPS, 08036 Barcelona, Spain; asolerp@clinic.cat (A.S.P.); veholguin@clinic.cat (V.H.A.); jrimola@clinic.cat (J.R.); joferrer@clinic.cat (J.F.-F.); magarcia@clinic.cat (Á.G.-C.); 3Centro de Investigación Biomédica en Red de Enfermedades Hepáticas y Digestivas (CIBERehd), 28029 Madrid, Spain; rgarciam@clinic.cat (R.G.); paruiz@clinic.cat (P.R.); elopezb@clinic.cat (E.L.); jcolme@clinic.cat (J.C.); 4Department of Medicine, Division of Hepatology, Centre Hospitalier de l’Université de Montréal, Montreal, QC H2X 3E4, Canada; gabrielle.jutras@umontreal.ca; 5Liver Transplant Unit, Liver Unit, ICMDM, Hospital Clinic Barcelona, IDIBAPS, University of Barcelona, 08007 Barcelona, Spain; 6Radiology Department, CDI, Hospital Clinic Barcelona, IDIBAPS, 08036 Barcelona, Spain; 7Hepatobiliopancreatic Surgery and Liver and Pancreatic Transplantation Unit, Department of Surgery, Hospital Clinic Barcelona, 08036 Barcelona, Spain; 8Universidad Jaume I, 12006 Castellón de la Plana, Spain; 9University of Barcelona, 08007 Barcelona, Spain; 10Departament of Medicine, Division of Gastroenterology and Hepatology, University of California—San Francisco, San Francisco, CA 94115, USA; jennifer.lai@ucsf.edu

**Keywords:** liver transplantation, hepatocellular carcinoma, immunotherapy, comorbidities, frailty, immunosuppression management, recurrence surveillance, multidisciplinary care

## Abstract

**Simple Summary:**

Liver transplantation (LT) is one of the primary treatments for hepatocellular carcinoma (HCC) and significantly improves patient survival. However, the application of LT for HCC faces challenges owing to advancements in cancer-specific treatments and increased patient comorbidities. This review explores the current controversies and advancements in LT for HCC, focusing on managing comorbidities, the impact of frailty, selection criteria, the role of radiology, and the potential use of immunotherapy. We emphasize the importance of immunosuppression management and surveillance for HCC recurrence. A multidisciplinary approach is crucial to optimize the outcomes of patients with HCC undergoing LT, ensuring comprehensive care and improved survival rates.

**Abstract:**

Liver transplantation (LT) remains one of the most effective treatments for hepatocellular carcinoma (HCC) and significantly enhances patient survival. However, the application of LT for HCC faces challenges owing to advancements in cancer-specific treatment modalities and the increased burden of patients’ comorbidities. This narrative review explores current controversies and advancements in LT for HCC. Key areas of focus include the management of comorbidities and patient education by advanced practice nurses, impacts of frailty on waitlists and post-LT outcomes, selection criteria for LT in the era of new downstaging tools, role of radiology in patient selection, and implications of potential immunotherapy use both before and after LT. Additionally, the importance of immunosuppression management with strategies aimed at minimizing rejection while considering the risk of HCC recurrence and the role of surveillance for HCC recurrence is highlighted. This review also underscores the importance of a multidisciplinary approach for optimizing outcomes in patients with HCC undergoing LT.

## 1. Introduction

Liver transplantation (LT) is one of the most beneficial treatments in terms of survival for patients with hepatocellular carcinoma (HCC) [1]. Globally, HCC is one of the primary indications for LT [2]. The applicability of LT in HCC has been constrained by donor availability and the need to ensure long-term survival benefits, thereby optimizing organ utility [3].

In recent decades, significant advancements have reshaped the LT landscape. These include the development of direct-acting antivirals (DAAs) [4], the expansion of the donor pool using living donors and donation after circulatory death (DCD) donors, and the advent of graft perfusion machines [5,6]. The indication for LT in HCC is increasingly characterized by an older, more comorbid population with a reduced functional reserve [7]. Furthermore, systemic treatments have evolved with the emergence of immunotherapies (IO), highlighting the potential roles of these treatments in the LT setting and presenting notable challenges in managing LT for HCC [8].

In this context, the present narrative review aims to examine key controversies in this field, providing a comprehensive overview of the evolving challenges and advancements in LT for HCC (Figure 1).

## 2. The Role of Advanced Practice Nurses in Patient Education and Managing Comorbidities

Currently, HCC and end-stage liver disease secondary to alcohol abuse and metabolic dysfunction-associated steatotic liver disease (MASLD) are the most common indications for LT. The increase in life expectancy in the last century has led to more elderly people (defined as over 65–70 years old) being considered for LT [9]. For these reasons, it is essential to perform a correct evaluation of the candidates, focusing on their current conditions, comorbidities, and previous health statuses [9,10]. Another important point is the assessment of alcohol, tobacco, and/or other drug use. This assessment should include a specific psychiatric assessment and objective abstinence tests [11,12,13]. The length of abstinence required depends on the protocols of each center. It is unknown whether active tobacco consumption should be a contraindication for LT. However, smoking, which can increase the chances of hepatic artery thrombosis post-LT, impacts the risk of cardiovascular and neoplasia as long-term complications, and it may be considered a contraindication to LT in patients with serious tobacco-associated comorbidities such as advanced vascular or lung disease or in those of older age (>70 years) [13]. Active smokers should be referred to specialized units for tobacco withdrawal [13,14].

Moreover, all transplant candidates need a psychosocial assessment including psychological status, resources for self-management, and family and social resources [15] to provide appropriate multidisciplinary interventions in this regard.

In elderly patients, the presence of comorbidities is more prevalent and may have a greater impact on both the patient’s physical condition and their quality of life [9,10]. Consequently, it is imperative to update general screening to detect possible neoplasms and perform an accurate cardiological evaluation, including the assessment of cardiovascular risk factors, (CVRFs) such as obesity, hypertension, hyperlipidemia and diabetes, while providing early and adequate education about these factors. This is particularly important considering that these CVRFs will be exacerbated by the side effects of immunosuppressants [16,17,18]. Other aspects to consider include frailty and neurological status. While frailty is specifically addressed in a different section of this review, neurological status should be evaluated using validated scales (the Mini-Mental State Examination and the Barthel Index) to detect a possible cognitive impairment [19].

Since the management of these patients is complex, a multidisciplinary approach is required, in which nurses have a specific key role as evaluators and educators [20,21,22]. The nurses involved must meet certain requirements, such as experience and knowledge of the pathology, and have the ability to respond to complex situations. For this reason, some centers have opted for the role of the advanced practice nurse (APN), who, in addition to providing expert care, develops other domains such as research, teaching, leadership, and evidence generation.

APNs accompany patients and caregivers from the detection of the disease to LT and adapt care planning to different stages of the process. The accompaniment of the APNs facilitates holistic care, addressing the physical, sociocultural, and emotional aspects in the following stages: (1) the detection phase [23], (2) the diagnosis phase [24], (3) the bridging treatment phase [1], (4) inclusion on the LT waiting list, (5) LT, and (6) post-LT care (Figure 2).

In all these interventions, active listening is key, and emotional support should be provided to help patients and their caregivers manage the distress generated by cancer diagnosis, time on the waiting list (WL), or the fear of transplant complications [25,26].

In summary, APNs play a specific role in the management of all phases of patients’ journeys, which are based on therapeutic education, counseling, and support. These interventions are targeted to empower patients to acquire knowledge, skills, and attitudes to effectively manage liver cancer and the LT process.

## 3. Prevalence and Impact of Frailty on Waitlists and Post-LT Outcomes in Patients with HCC

Frailty is defined as a “distinct biologic state of decreased physiologic reserve and increased vulnerability to health stressors” [27]. This state results from the accumulation of age-related deficits across various domains, including psychological, physiological, social, and cognitive ones [28,29]. Although originally conceptualized in the field of geriatrics, frailty is now well recognized within the LT community as a construct that can facilitate comprehensive assessment of a liver patient’s overall vulnerability to adverse transplant-related outcomes, extending beyond the risks associated with end-stage liver disease alone.

The Liver Frailty Index (LFI), a standardized frailty assessment tool developed specifically for patients with end-stage liver disease, includes three performance-based tests that capture key domains of frailty predominantly affected in patients with end-stage liver disease [30]. This tool has proven to be reliable and reproducible when administered to LT candidates, thereby standardizing the evaluation of frailty and comparison of its prevalence in this population [31]. Research using the LFI in LT candidates demonstrates a high prevalence of frailty, ranging from 15 to 43% [30,32]. In this population, frailty is more prevalent among older adults and women [33,34]. Patients with MASLD also display higher rates of frailty, reflecting higher rates of metabolic comorbidities [35]. Not unexpectedly, rates of frailty increase with liver disease severity, as reflected by the Model for End-Stage Liver Disease (MELD) score, ascites, hepatic encephalopathy, and the Baveno classification [36,37].

Frailty has been strongly linked with poor outcomes both before and after LT. In a study of over 1000 ambulatory LT candidates with cirrhosis in the United States (U.S.), frailty was associated with a nearly 2-fold increased adjusted risk of death [38]. Changes in frailty over time, whether worsening or improving, are also informative of mortality risk. A multicenter study in the U.S. evaluating >1000 patients with cirrhosis demonstrated that each 0.1-unit change in the LFI over 3 months was associated with a 2-fold increased risk of WL mortality, independent of the MELD-Na score [38]. Post-LT compromised functional performance and pre-LT frailty were found to be associated with higher hazard ratios for death [33,39,40].

Beyond mortality, baseline frailty measures have also been linked with important adverse healthcare outcomes post-LT. These include increased post-LT healthcare costs, prolonged hospital stays nearly twice as long as those of non-frail counterparts, a higher frequency of discharge to rehabilitation facilities (versus home), and increased readmissions [41,42]. Frailty is also strongly associated with poorer patient-reported outcomes, such as more frequent falls and lower health-related quality of life [43,44]. Importantly, frailty has been identified as a potential target for improving prognosis, particularly by identifying components that can be addressed during the pre-transplant rehabilitation period.

Patients with HCC form a distinct subgroup of LT candidates with unique characteristics that necessitate careful consideration when it comes to clinical applications of the frailty construct. These patients receive exception points for prioritization on the WL that do not reflect their underlying hepatic dysfunction. Consequently, they may represent a diverse subpopulation with varying degrees of illness and impaired physical status. While LT candidates with HCC tend to be older than LT candidates without HCC and therefore may have a higher burden of non-hepatic comorbidities that would contribute to frailty, they often have less advanced synthetic dysfunction and portal hypertensive complications. While underlying malignancy and treatment side effects are known to contribute to frailty in the general population, HCC itself is often asymptomatic, and locoregional therapies are well tolerated.

Few studies have purposely examined the prevalence of frailty in LT candidates diagnosed with HCC. One initial study, which included 50 LT candidates with HCC, reported that 30% met the criteria for frailty using the Fried Frail Phenotype (FFP) [42]. A larger U.S. multicenter study that included 501 patients with HCC observed strikingly similar results, with 27% demonstrating frailty using the LFI [40]. These results indicate that LT candidates with HCC are at least as frail as their non-HCC counterparts.

Research on the impact of frailty on LT-related outcomes in this subgroup is also limited. However, existing studies show that frailty in LT recipients with HCC is associated with worst post-LT outcomes, similar to those seen in non-HCC candidates. These include notably prolonged hospital stays and delayed physical therapy initiation [42]. In patients with chronic liver disease with HCC, frailty has also been linked to lower survival rates and other adverse health outcomes. A retrospective cohort analysis using the Nationwide Inpatient Sample (NIS) found that frailty increased the risk of inpatient death by 4.5 times and the likelihood of developing hepatic encephalopathy during hospitalization by 2.3 times [42,45]. Similarly, frail older patients undergoing hepatectomy for HCC have significantly lower 3-year cancer-specific survival rates and 5-year overall survival (OS) rates after liver resection compared to non-frail patients [46,47]. These frail patients also required more healthcare support after discharge, highlighting the additional challenges that they face in recovery [47]. Beyond mortality, frailty is associated with poorer physical health in patients with chronic liver disease and HCC. Specifically, frailty, as measured by the LFI, is negatively correlated with sarcopenia and independently predicts muscle atrophy in these patients [48].

Given the significant role that frailty plays in adverse outcomes for patients with HCC, interventions targeting frailty would theoretically offer substantial benefits in improving patient outcomes. Accordingly, a multicenter observational study highlighted the efficacy of an in-hospital exercise program in significantly improving frailty in HCC patients [49]. Ongoing research is currently exploring the feasibility and acceptability of home-based exercise sessions for patients with HCC, aimed at improving frailty and overall health outcomes in this cohort [50].

In conclusion, frailty is as prevalent in LT candidates with HCC as in those without HCC. Like the non-HCC population, frailty is strongly associated with adverse health outcomes in HCC patients. Therefore, frailty measurement is an essential factor in patients with HCC awaiting LT as it provides a comprehensive assessment of the potential combined effect of underlying liver dysfunction, as HCC, along with advancing age and comorbidities, is very common in HCC patients. Routine assessment of HCC could help to identify higher-risk patients and inform the development of tailored interventions to mitigate these risks. By better understanding frailty prevalence and its implications for HCC LT candidates, strategies can be developed to optimize outcomes, ultimately enhancing quality of life and long-term survival for these patients.

## 4. Selection Criteria for LT in HCC: Should Downstaging Be Universal or Tailored to the Tumor Burden and AFP Limits?

The Milan criteria (MC) [51], defined as the presence of a single tumor smaller than 5 cm or up to three tumors each smaller than 3 cm, with the absence of macrovascular invasion or extrahepatic spread, have been, for many years, the cornerstone selection criteria for LT in HCC. However, several researchers have argued that these criteria are too restrictive and exclude certain patient subgroups who might otherwise benefit from LT. Consequently, several expanded criteria have been proposed. The application of these less stringent criteria could potentially increase the pool of LT candidates by 5–10% without adversely affecting survival rates [52]. While most expanded criteria are based on the number and size of tumor nodules [53], the incorporation of biomarkers as surrogates for biological behavior has gained significant interest, particularly for alpha-fetoprotein (AFP). AFP levels have been demonstrated to be highly predictive of patient survival [54], with the AFP Model [55] and Metroticket 2.0 [56] being among the most validated criteria. Expanding criteria in terms of tumor burden entails a higher risk, and in this scenario, downstaging (DS) has been introduced with the aim of reducing tumor burden to select a subgroup of patients with favorable tumor biologies and prognoses that align with the accepted criteria for LT. The efficacy of DS is supported by a recent randomized controlled trial, in which this strategy demonstrated a clear survival benefit with a 5-year OS rate of 77.5% [57]. Although transarterial chemoembolization (TACE) is the most frequently used treatment for achieving DS [58], the use of alternative therapies, including other locoregional (radiofrequency ablation, microwave ablation or radioembolization), systemic treatments, or a combination thereof, has gained significant interest recently. Nevertheless, the safety of patients, both before and after LT, remains an essential consideration in any DS protocol under evaluation for implementation.

Although DS is widely accepted as a method for expanding LT indication in HCC, unresolved issues remain, such as establishing the limits on tumor burden and AFP levels for DS candidacy, as well as defining the criteria for successful DS in terms of the extent and duration of tumor response. To standardize DS criteria, the United Network for Organ Sharing (UNOS) implemented the UCSF/Region 5 DS protocol as a national policy in the United States in 2017 (UNOS-DS protocol), which includes the following: a single lesion measuring 5.1–8 cm; two to three lesions, each smaller than 5 cm; and criteria for four to five lesions. It also stipulates that patients with elevated AFP, even those exceeding 1000 ng/mL, may still be considered for inclusion if their AFP levels decrease below 500 ng/mL and their disease remains stable after DS for a period of six months [21,59]. However, based on external validations in the USA, an AFP < 100 ng/mL is recommended to optimize candidate eligibility [54].

### Is the “All Comers” Approach Worth It?

Major challenges are to delineate the upper limits for attempting DS and to determine the threshold beyond which DS is considered infeasible (colloquially referred as “how much is too much”). The lack of a standardized DS consensus protocol is partially justified by the global variations in WL dynamics and donor availability. Furthermore, it is imperative that the criteria for staging both before and after treatment are standardized to ensure that outcomes are generalizable across a wider population and aligned with the intention-to-treat (ITT) principle. In response to this challenge, the ‘all-comers’ (AC) strategy has been proposed to broaden LT eligibility to patients initially exceeding the expanded criteria, with the potential to meet the required tumor reduction. However, to understand the true impact of this approach, it is necessary to analyze it using the ITT principle. The key questions that need addressing are how efficient and safe is this approach, and does it provide better survival outcomes for patients compared to other treatments supported by scientific evidence? In this context, Sinha et al. showed that patients in the AC group exhibited a lower success rate of tumor DS, an increased cumulative probability of dropout from the WL, and a lower 5-year ITT survival rate compared to those who complied with the UCSF-DS criteria (50% in the AC group vs. 78.5% in the UCSF-DS group) [60]. Furthermore, Mehta et al. demonstrated that the 3-year survival rates after LT were comparable between patients who were successfully downstaged to meet the UNOS-DS criteria (79%) and those within MC (83%). However, the 3-year survival rate after LT for the AC cohort was significantly lower than that of other two groups [54]. Therefore, current research indicates worse outcomes in AC patients compared to patients who meet the UCSF-DS criteria [61].

As previously exposed, the current landscape permits the expansion of LT criteria for HCC. However, it is imperative to strive for competitive patient survival outcomes. This necessitates a meticulous selection process for HCC patients, which involves not only the precise evaluation of tumor burden through accurate and correctly interpreted imaging tests but also an assessment of the tumor’s biological behavior [62]. Such careful selection is essential to minimize the risk of patient dropout due to tumor progression and subsequent removal from the WL, while also preserving the integrity of organ allocation for non-HCC patients. To address unresolved questions related to the acceptable survival rate threshold and aid decision-making processes, a thorough evaluation is indispensable. This analysis should incorporate the considerations of maximizing community benefit (utility), equitable distribution of those benefits (distributive justice), respect for patient autonomy, and adherence to the principle of beneficence. Adopting this multifaceted approach ensures that decisions are made with an informed and balanced perspective on survival rates and other pivotal considerations.

## 5. The Pivotal Role of Radiology in Patient Selection for LT

Given that HCC can be diagnosed noninvasively by typical imaging features, accurate imaging diagnosis and staging are essential for determining transplant eligibility. The diagnosis of HCC typically relies on identifying characteristic imaging features using computed tomography (CT) and magnetic resonance imaging (MRI). The Liver Imaging Reporting and Data System (LI-RADS) diagnostic algorithm, introduced in 2011 and later enhanced by liver-specific hepatobiliary contrast agents in 2014, aims to standardize the interpretation and reporting of liver imaging findings. LI-RADS was fully integrated with the AASLD HCC clinical practice guidelines in 2018 [63] and the most recent 2023 version [21].

Although LI-RADS criteria are widely accepted for the initial diagnosis of HCC, their integration into pre-LT staging and tumor burden assessment remains challenging. The high-risk stages of HCC represent a significant likelihood of malignancy, complicating the application of the LI-RADS in stratifying and evaluating tumor burden within the LT context. A key question is whether to include LR-3, LR-4, or LR-M in assessing tumor burden. Some studies suggest that excluding LR-3 and LR-4 reduces the accuracy of tools such as Metroticket 2.0, which predicts post-LT outcomes, while others indicate that including LR-4 and LR-M alongside LR-5 provides similar accuracy for LT eligibility based on the Milan criteria [64].

Another crucial aspect to consider when applying noninvasive diagnostic criteria is that these criteria have been validated at the level of a single lesion. However, they have not considered the potential interaction that can arise from the coexistence of different focal lesions in the same liver, particularly when two or more focal lesions coexist, in which at least one does not meet the conclusive criteria for HCC.

Gadoxetic acid is a liver-specific MRI contrast agent that provides hepatobiliary phase imaging reflecting hepatocyte-specific uptake. In contrast to the liver parenchyma, focal liver lesions that do not contain functioning hepatocytes, such as most HCCs, display an absence of uptake in the hepatobiliary phase. This finding is regarded as an ancillary for the diagnosis of HCC or malignant lesions in the LI-RADS system.

However, the diagnostic performance of gadoxetic acid for staging HCC in LT candidates varies among studies, with a sensitivity ranging from 35 to 61% [65,66]. The efficacy of liver-specific MRI, particularly in the identification of indeterminate nodules, as well as the contribution of the hepatobiliary phase to improving the staging for the assessment of LT eligibility, remains uncertain.

Because of these limitations, clinical guidelines do not reflect how to handle indeterminate nodules with atypical enhancement features [21,67]. Deciding between follow-up imaging and an intense work-up, including alternative imaging techniques or biopsy, remains controversial. A recent study revealed that >60% of indeterminate nodules, mostly <3 cm, identified by dynamic MRI were pathologically confirmed as HCCs, regardless of size [68]. These findings impacted the eligibility criteria for LT both before and after the procedure. Additionally, HCC cells were found microscopically in 35% of indeterminate nodules categorized as LR-2 or LR-3 on MRI, and nearly 70% were categorized as LR-4 lesions. Therefore, rigorous follow-up testing is recommended for LR-2 and LR-3 lesions deemed probably benign or indeterminate [68]. This suggests that the adoption of the LI-RADS for liver organ allocation might have inherent biases.

Despite extensive research on contrast-enhanced ultrasound (CEUS) for liver imaging, the AASLD and EASL guidelines still do not recommend it as a first-line modality for HCC diagnosis [21,67]. However, a recent study confirmed the high diagnostic performance of CEUS LR-5, with 95.1% specificity and 97.3% positive predictive value for HCC. It has also been demonstrated that LR-5 can effectively exclude most non-HCC malignant nodules, such as intrahepatic cholangiocarcinoma (ICC), combined ICC-HCC, and metastases [69]. This diagnostic performance is similar to the results observed for CT/MRI LI-RADS [70]. Additionally, it offers a reliable and safe imaging method for characterizing liver nodules in patients who cannot undergo CT or MRI. Causal issues include renal insufficiency, allergies to iodinated or gadolinium-based contrast agents, claustrophobia, or the presence of metallic implants and foreign bodies. The primary controversy surrounding the use of solely US coupled with CEUS is the potential understaging of HCC. Nevertheless, the high specificity of CEUS for characterizing MRI-indeterminate nodules indicates that the selective application of CEUS to non-LIRADS-5 nodules on MRI or CT performed in candidates for LT could be an effective strategy for staging optimization. In recent years, artificial intelligence (AI) has seen significant growth in radiology, utilizing machine learning and deep learning techniques. These advancements in AI have the potential to revolutionize LT management by improving therapeutic decisions and patient selection. However, the variability in and lack of reproducibility of many studies have limited their clinical impact so far [71]. Nonetheless, AI models are expected to become valuable tools for LT management in the future.

Finally, the continuous evolution of imaging techniques and criteria development plays a critical role in ensuring that LT remains an effective treatment for these patients, especially if the goal is to expand access by extending the classic limits of transplant eligibility. This is particularly important given the complex challenges posed by organ scarcity.

## 6. Immunotherapy Pre- and Post-LT: Expanding Horizons and Challenges

The advent of immunotherapy in oncology (IO) has shaped the landscapes of most cancers. This is also the case for HCC, where four immunotherapy-based regimens have been proven to be superior in first line in terms of OS compared to the standard of care (sorafenib or lenvatinib) [72,73,74,75]. The unprecedented survival data are accompanied by very high rates of radiological response, ranging between 14.3% and 42%, depending on the regimen and criteria of response (Table 1). Thus, the attempt to evaluate the potential benefit of IO in earlier stages than in the advanced stage seems reasonable. Together with neoadjuvant and/or adjuvant and combination therapies, the assessment of IO in the LT field is ongoing. Nevertheless, the latter setting is a more complex environment where the evaluation of the benefit of IO should focus on efficacy not only at the individual level but also at the population level [3]. Additionally, the safety concern is a crucial issue in transplant oncology because the targets of the most commonly used immune checkpoint inhibitor (ICI) regimens in HCC (PD-1/PDL-1 and CTLA-4) are also involved in the induction and maintenance of solid organ tolerance [76]. The potential applications of IO in this setting can be distinguished in the pre-LT (both downstaging and bridging therapy) and post-LT scenarios and will be analyzed separately.

### 6.1. Pre-LT: Downstaging and Bridging Therapy

The initial and absolute skepticism of IO safety in LT candidates after the first fatal reports has been progressively counterbalanced by case reports and series reporting lower rates of liver rejection than previous cases (Appendix A). Up to June 2024, 158 patients were treated with IO before LT, according to the published studies. Tabrizian et al. reported the experiences of nine patients who received LT after ICI monotherapy, with only one developing an episode of mild acute rejection [77]. A retrospective multicenter study from China reported a 27.7% rate of rejection in patients previously treated with a variety of ICIs. In the same study, an interval shorter than 30 days between the last dose of ICI and LT was identified as a predictor of rejection [78]. Interestingly, the mortality rate of patients who developed rejection was around 26% (6 out 23 patients). In a multiregional and retrospective study from the U.S. including 30 patients treated with ICIs, the pre-LT rejection rate at 1 year was 16.6%, and patients with a washout period longer than 90 days had a significantly lower probability of rejection. Although ICIs usually have a half-life of <28 days, PD-1 occupancy in lymphocytes has been observed for up to 100 days [79]. Similarly, a recent meta-analysis of 91 patients showed that older age and a longer ICIs washout period are significantly associated with a reduced risk of allograft rejection. Specifically, each 10-year increase in age decreases the risk by 28%, and each additional week in the washout period reduces the risk by 8% [80]. In addition, it is important to emphasize the need to consider blood product transfusions and estimated blood loss during LT surgery when calculating the probability of acute rejection in patients who have used ICIs pre-LT. These factors may influence the pharmacokinetics of ICIs and, consequently, the risk of graft rejection. Moreover, peri-LT plasmapheresis could hypothetically play a role in mitigating the risk of rejection, especially in cases with minimal intra-operative blood loss and a short duration of ICI treatment [81]. However, while the results are promising, the absence of data on the type of immunosuppressive regimen and the cumulative dose of calcineurin inhibitors in the analysis makes the interpretation challenging, as these variables are crucial in terms of the risk of graft rejection. This is a crucial piece of the story that is required to globally evaluate the safety of IO before LT, in addition to the mere rate of rejection. Cumulative exposure to tacrolimus predicts the rate of post-LT de novo malignancies [82]. Similarly, high doses of immunosuppressive regimens can also affect other long-term safety outcomes such as metabolic disorders and nephrological complications. Therefore, the results of ongoing clinical trials in the field should focus on this critical aspect. Nowadays, ten clinical trials are ongoing (NCT04425226, NCT05185505, NCT05027425, NCT04443322, NCT04035876, NCT05411926, NCT05475613, NCT05339581, NCT05913583, and NCT05879328), but none of them mention the potential side effects of highly immunosuppressive regimens among the secondary endpoints. Among these trials, the interim analysis of the PLENTY (NCT04425226) was presented at ILC-2023, and 22 patients with HCC beyond Milan were randomized to pembrolizumab in combination with lenvatinib versus routine treatments before LT. Here, no acute allograft rejection was reported, and recurrence free survival (RFS) at 1 year was 62.5% and 37% in the control arm. These results suggest that a significant proportion of patients were not suitable candidates for LT. The immuno-XXL trial is an observational single-arm study including patients on the WL after the achieving OR by mRECIST, lasting at least three months with atezolizumab plus bevacizumab at the standard dose with RFS as the primary endpoint. The preliminary results of the ten patients who underwent transplanted were reported after a median follow-up of six months (Bohori S et al. ILC-2024); one patient suffered liver rejection, and no tumor recurrence was registered. As for PLENTY, data from the immune-XXL are still immature, and long-term results in terms of safety and efficacy are awaited.

Concerning the objective of IO therapies prior to LT, both the bridging and the downstaging scenarios are currently being assessed. Nevertheless, prior to the publication of compelling safety data, the use of IO therapy as a bridging strategy must be considered within the context of clinical trials or research studies. In fact, the former approach implies that a patient is already a candidate for LT and that the benefits of treatment could vanish if the short- and long-term risks are relatively high. Consequently, in this particular context, it is recommended to opt for already available locoregional treatments. However, it must also be acknowledged that the combination of IO with locoregional treatments can result in an increase in radiological response. This is exemplified by the phase 3 EMERALD-1 trial, in which the combination of TACE with durvalumab and bevacizumab demonstrated improved PFS, time to progression, and OR compared with TACE alone [83]. However, mature data on OS and the declaration of the reason for censoring will be crucial to understand the true benefits and potential applications of this combination. It is important to highlight that there are controversies in the assessment of the radiological response depending on the criteria used. The mRECIST criteria may overestimate the radiological response and underestimate disease progression (e.g., new lesions), as they are not validated in the context of IO. In contrast, while RECIST 1.1 criteria may be considered more reliable, they can be slower to register a response.

A crucial aspect to remember, especially in the context of downstaging, is that although a positive response is undeniably beneficial, it does not necessarily indicate that the best option for the patient is surgery, such as liver resection or LT. Indeed, studies directly comparing the efficacy of continuing immunotherapy versus surgical treatments once OR is achieved are still lacking. For this, it is worth recalling that once response is achieved with immunotherapy, the duration of response (DOR) is long and ranges between a median of 14.8 and 30.4 months (Table 1). Additionally, the updated analysis of the HIMALAYA trial showed that among long-term survivors (>36 months), 51.5% had an OR as determined by RECIST v1.1 [84]. Therefore, the assessment of the benefit of LT after IO should be compared according to the intention-to-treat analysis of the benefit of IO alone.

In summary, IO before LT is a flourishing field of research but cannot be endorsed for practical application yet. The results from ongoing clinical trials are largely awaited, but a thorough evaluation of the survival benefit, together with the overall safety profiles in the short and long term, is mandated. Finally, it is crucial to consider the availability of donors and WLs for LT dynamics in the specific region under evaluation in order to assess the applicability and potential impact of this expansion of criteria.

### 6.2. Implications of Immunotherapy Use in the Post-LT Setting

The post-LT setting is characterized by the risk of rejection and, more importantly, graft loss. This aligns with evidence from the initial upsetting results in HCC with reported graft losses of up to 50% (Appendix A). A recent systematic review of 31 case reports, including 52 patients who were treated with IO for de novo malignancies after LT, reported a rejection rate of 28.8% [85]. Not surprisingly, rejection was associated with a shorter OS. However, all these data coming from case reports should be cautiously interpreted considering the evident publication bias. In this vein, a multicenter and retrospective French study that included 35 patients with LT treated with IOs for de novo or recurrent HCC reported 17% acute rejection, but the condition was fatal in 66.6% of them (De Martin E et al. ILC-2024).

As of now, only two clinical trials testing the safety of IO in patients undergoing LT are ongoing, namely NCT03966209 and NCT06254248. Conversely, treatment with tyrosine kinase inhibitors (TKI) has demonstrated an acceptable safety profile and acceptable efficacy data in the treatment of HCC recurrence post-LT [21].

Currently, there is no evidence to support the use of IO in patients with prior LT. The recommendation for advanced recurrences remains the use of TKI. The use of IO in this context, and outside of clinical trials, might be considered a salvage option for patients who have exhausted all therapeutic options. This approach mandates specific surveillance and immunosuppression strategies, along with clear medical–patient consensus regarding the potential risks and benefits.

## 7. Immunosuppression Management of LT in HCC

Advances in immunosuppression during the initial decades of LT have significantly improved graft and recipient survival. Calcineurin inhibitors are the cornerstones of rejection prevention and treatment [86]. Tacrolimus is preferred over cyclosporine due to its lower rejection and mortality rates [87]. Given its narrow therapeutic range, monitoring plasma tacrolimus levels is warranted to prevent rejection but avoid side effects [88]. Some of these may impair recipient morbidity and prognosis, such as renal injury, increased incidence of cardiovascular complications, or higher risk of neoplasia. This increased risk of cancer may be associated with de novo neoplasia (solid organ or hematological) and HCC recurrence, and it seems to be associated with a higher cumulative exposure to calcineurin inhibitors [82,89]. Moreover, tacrolimus trough levels during the first month after LT seem to have a major impact on the HCC recurrence rate, especially those patients with plasma levels higher than 10 ng/mL [90,91,92]. Thus, patients transplanted for HCC may benefit from lower tacrolimus target levels (6–10 ng/mL) during the first month and even lower levels (4–6 ng/mL) during the maintenance phase [18].

A frequent strategy to minimize tacrolimus doses without increasing the risk of rejection is to combine tacrolimus with other immunosuppressants. A reduction in tacrolimus exposure with the addition of a second drug has been associated with a significant improvement in renal function in the first year and following years after LT [93,94]. Similarly, this reduction in tacrolimus exposure may result in a lower HCC recurrence rate. Among the immunosuppressants to be combined, the mammalian targets of rapamycin (mTOR) inhibitors, everolimus and sirolimus, have been suggested to be the most suitable because of their antiproliferative properties [95]. Previous studies have shown that the combination of tacrolimus and mTOR inhibitors is safe and associated with a significantly better glomerular filtration rate during the first two years after LT [94,96,97], but its role in the prevention of HCC recurrence is controversial. Although retrospective studies and meta-analyses have claimed a favorable profile for these drugs [98], clinical trials have failed to show the benefits of this combination for HCC recurrence rates [99,100]. Likewise, the combination of tacrolimus and mycophenolate has shown contrasting results [101]. Current evidence does not support the use of one or other combinations for HCC recurrence prevention. Nevertheless, for patients with a high risk of recurrence (HCC out of the Milan criteria, histological findings such as microvascular invasion or satellitosis), tacrolimus with reduced target levels combined with mTOR inhibitors may be beneficial. However, the minimization of tacrolimus doses and trough level targets must be carefully assessed, as it could result in an increased risk of de novo donor-specific antibodies and rejection [102,103].

When HCC recurrence is already diagnosed, immunosuppressive treatment should be cautiously reassessed regarding cancer-specific treatment options. In general, patients with cancer after LT are usually subjected to either immunosuppressive drug minimization or a switch to or combination with mTOR inhibitors, which has been proven to have better outcomes in certain types of neoplasia. However, there are no current recommendations or protocols for this minimization in the setting of HCC recurrence due to a lack of evidence of the impacts of these approaches on HCC response and patient survival [13,104,105]. The antiproliferative property of mTOR inhibitors has also been claimed to justify its synergistic use in patients receiving sorafenib as a systemic treatment for disseminated recurrent HCC. However, the contrasting results of the reported evidence refrain from recommending a systematic switch to this immunosuppressant in these patients [106].

## 8. Post-LT Surveillance for HCC Recurrence

HCC recurrence most frequently appears within the first 2 years post-LT. However, late recurrences (>2 years after LT) can also occur, although they are less common beyond 5 years after LT [107]. Post-LT HCC recurrence can manifest as disseminated disease, oligo-recurrence, or solitary lesions. HCC recurrence may occur in the liver, extrahepatic sites, or both [108]. The lungs and bones are the most common extrahepatic sites [105]. Early disseminated recurrences usually exhibit a worse prognosis than late extrahepatic oligo-recurrences. The current prognosis of disseminated HCC recurrence after LT is poor. In contrast, the surgical resection of recurrent HCC, particularly solitary lesions, is associated with enhanced long-term survival [105]. The detection of solitary or early oligo-recurrences after LT using CT surveillance may increase the likelihood of undergoing aggressive treatments, thereby improving patient survival [109]. This evidence partly supports the implementation of surveillance for HCC recurrence after LT [110,111]. However, the retrospective nature of these studies, associated with selection biases and inconsistent data collection, limits the ability to establish definitive causal relationships. Additionally, the lack of a standardized definition of “evaluation intervals” may lead to varied interpretations and applications, further complicating the generalization of the findings.

### 8.1. Selection of Candidates for Surveillance

The debate between monitoring all patients and only those at high risk of HCC recurrence remains open. Several variables have been associated with HCC recurrence after LT. AFP levels before LT, the number and size of nodules, poor differentiation of the tumor, and the presence of microvascular invasion (mVI) or satellitosis on explant are strongly associated with the risk of HCC recurrence [112,113]. Several prediction models use these variables to identify patients with a higher risk of recurrence before and after LT. Among them, Metroticket 2.0, the AFP model (before LT), the MORAL score, RELAPSE, HALT-HCC, LiTES-HCC or R3-AFP, and the RETREAT score (after LT) have been described in large series of patients [114,115]. The RETREAT score, which is calculated based on the AFP levels at transplantation, mVI, and the largest viable tumor on explant, has been extensively validated [114].

Despite the ongoing debate, recent consensus, such as ILCA-ILTS 2024 or the new EASL guidelines of LT 2024, recommend surveillance for intermediate- to high-risk individuals.

### 8.2. Duration, Intervals, and Imaging Techniques for Surveillance of HCC Recurrence after LT

Successful surveillance should not only focus on patient risk but should also target the timeframe when recurrences are most likely to occur, assess the sites with higher chances of recurrence, and employ imaging techniques with high sensitivity and specificity. Due to the non-linear incidence of HCC after LT, shorter screening intervals for early HCC recurrence detection have been reported. The most widely recommended approach is to perform surveillance every 6 months. This interval has been associated with the possibility of applying curative treatment to recurrences [109]. Some authors suggest a follow-up every 3–4 months, initially for patients at very high risk [114]. It should be continued for at least 2–3 years post-transplant as most patients’ illnesses recur at this time. Most programs continue surveillance until five years after LT based on the low likelihood of HCC recurrence. The preferred technique to detect hepatic and extrahepatic recurrences is a thoraco-abdominal-pelvic CT scan. There are limited data regarding the efficacy and cost-effectiveness of AFP assessment in post-LT patients, yet some authors recommend monitoring AFP levels every six months for a period of five years [114]. Currently, there are not enough data to support the use of alternative biomarkers.

The impact of the implementation of well-defined surveillance protocols in the coming years could dictate future standards for post-LT management. However, methodological limitations undermine the robustness of recommending intensive post-LT screening, and such recommendations should be approached with caution. Current evidence, based on retrospective studies with variability between centers and non-standardized definitions, does not provide a solid foundation for universal screening recommendations. Nonetheless, it does suggest a potential benefit in a disease that is typically diagnosed at advanced stages. Therefore, additional research, preferably prospective studies, is needed to better define surveillance protocols and their impacts on post-LT survival.

## 9. Conclusions

The landscape of LT for HCC has markedly evolved in recent years. The criteria for LT candidacy have been extended, recognizing the value of DS as an effective method for identifying patients who are likely to achieve more favorable post-LT outcomes. This expansion also includes the consideration of older patients, who more frequently have frailty and additional comorbidities. Furthermore, the integration of IO into therapeutic options has radically transformed the approach to caring for HCC patients. Considering these advancements, a comprehensive and multifaceted assessment of candidates for liver transplantation is essential to optimize the advantages and implement the procedure effectively for the modern therapeutic management of HCC. Addressing the challenges highlighted in this review is crucial, as overcoming them could significantly enhance the success of liver transplantation in HCC patients, ultimately improving patient outcomes and advancing the field.

## Figures and Tables

**Figure 1 cancers-16-03059-f001:**
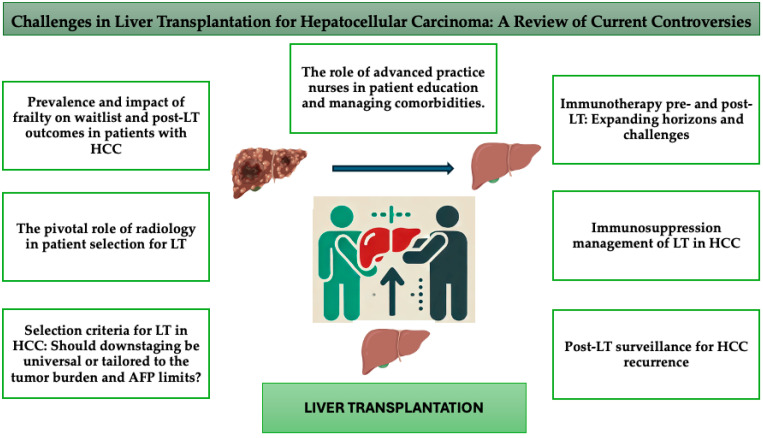
Challenges and key areas in liver transplantation for HCC.

**Figure 2 cancers-16-03059-f002:**
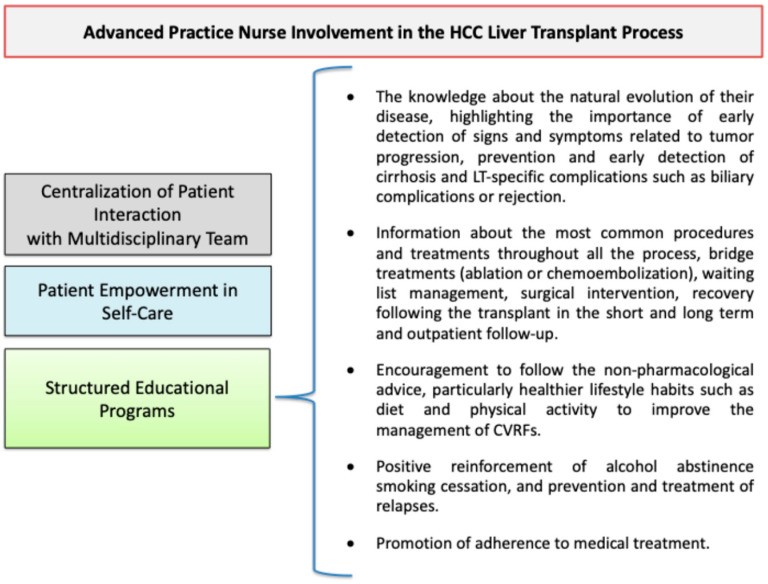
Comprehensive APN support throughout the HCC liver transplant journey.

**Table 1 cancers-16-03059-t001:** Objective response rate and duration of response in positive phase III clinical trials including ICI in first line setting.

Combination	Objective Response (RECIST v1.1), %, (95% CI)	Objective Response (mRECIST), %, (95% CI)	Duration of Response Months, Median (IQR)
Atezolizumab–Bevacizumab (IMbrave150)	30 (25–35)	33.2 (28.1–38.6)	18.1 (4.6–NE)
Tremelimumab–Durvalumab (HIMALAYA)	20 (NA)	NA	22.3 (8.5–NE)
Camrelizumab–Rivoceranib (CARES-310)	26 (20–35)	NA	14.8 (8.4–NE)
Durvalumab (HIMALAYA)	17	NA	16.8 (7.4–NE)
Tislelizumab (RATIONALE-301)	14.3 (10.8–18.5)	NA	36.1 (16.8–NE)
Ipilimumab–Nivolumab (Checkmate 9DW)	36 (31–42)	NA	30.4 (21.2–NE)

CI: Confidential Interval; NA: Not Available.

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
