# Peer review of "Challenges in Liver Transplantation for Hepatocellular Carcinoma: A Review of Current Controversies"

_cancers, 2024, doi:10.3390/cancers16173059_

Round 1

Reviewer 1 Report

Comments and Suggestions for Authors

Dear Authors, 

Your paper provides a a broad review about liver transplant and HCC, however it would be helpful to add flow charts or statements as take home messages. 

Other comments:

-It is not mentioned the role of the advance practitioner nurses in the follow up post liver transplant to ensure a close follow up with images. Also this role is not widely recognised in different countries.

-It is not mentioned that smoking can increase the chances of hepatic artery thrombosis post liver transplant. 

Author Response

Reviewer #1: Your paper provides a broad review about liver transplant and HCC, however it would be helpful to add flow charts or statements as take home messages. 

Other comments:

-It is not mentioned the role of the advance practitioner nurses in the follow up post liver transplant to ensure a close follow up with images. Also this role is not widely recognised in different countries.

-It is not mentioned that smoking can increase the chances of hepatic artery thrombosis post liver transplant. 

R: Thank you very much for your comments. We have emphasized the role of Advanced Practice Nurses (APNs) in post-LT follow-up within the manuscript, and we believe the newly added Figure 2 further highlights their role in this context. Lastly, we have incorporated the concept of smoking as a risk factor for hepatic artery thrombosis in the liver transplant setting. Regarding the suggestion to add a section with take-home messages, we have chosen not to include it, as we have already synthesized the key challenges explored in the Simple Summary section. Many of the topics discussed are subjects of ongoing debate and continuous information generation.

Reviewer 2 Report

Comments and Suggestions for Authors

The review article entitled 'Challenges in Liver Transplantation for Hepatocellular Carcinoma: A Review of Current Controversies' by Ezequiel Mauro et al., highlights the challenges, current controversies, and advancement in liver transplantation for hepatocellular carcinoma. The article is well written with some minor typo errors. The manuscript can be improved by addressing the details below. 

1. The Correspondence address is truncated.

2. Line 103 to 127, Page 3 is confusing. The message can be conveyed better using a schematic image/flow diagram.

3. The availability of donors for liver transplant is a major concern in terminal hepatic diseases cure. The review focuses on discussing/improving HCC therapeutic management including post-LT surveillance. I am wondering it would be great to include any latest studies on any constant feedback/loop system eg. through APN to facilitate timely liver availability/transplantation in HCC patients. 

Comments on the Quality of English Language

Overall the article is well written and arranged. However, the language can be simplified with figures/tables for conveying the ideas quicker. 

Author Response

Reviewer #2: The review article entitled 'Challenges in Liver Transplantation for Hepatocellular Carcinoma: A Review of Current Controversies' by Ezequiel Mauro et al., highlights the challenges, current controversies, and advancement in liver transplantation for hepatocellular carcinoma. The article is well written with some minor typo errors. The manuscript can be improved by addressing the details below. 

  1. The Correspondence address is truncated.
  2. Line 103 to 127, Page 3 is confusing. The message can be conveyed better using a schematic image/flow diagram.
  3. The availability of donors for liver transplant is a major concern in terminal hepatic diseases cure. The review focuses on discussing/improving HCC therapeutic management including post-LT surveillance. I am wondering it would be great to include any latest studies on any constant feedback/loop system eg. through APN to facilitate timely liver availability/transplantation in HCC patients. 
  4. Overall the article is well written and arranged. However, the language can be simplified with figures/tables for conveying the ideas quicker.

R: Thank you for your review and suggestions.

  1. We have updated the correspondence address as requested.
  2. We have adopted the suggestion to include Figure 2, which schematically simplifies the message.
  3. We appreciate your suggestion, which is relevant to the liver transplant scenario. However, given the complexity of the topics discussed, we believe that addressing additional themes might detract from the central focus of this study. Moreover, this particular topic is highly dependent on the organ availability rates in each region; we are grateful for your suggestion.
  4. Please, see point 2.

Reviewer 3 Report

Comments and Suggestions for Authors

This is a nice review manuscript on the challenges of liver transplantation for hepatocellular carcinoma, covering many topics, including the use of immunotherapy for immunosuppression. Here are some comments.

1. It will be nice to include a section on why liver transplantation is one of the primary treatment of hepatocellular carcinoma rather than other therapies such as chemotherapy.

2. The authors could better explain why challenges they choose to highlight are essential and whether overcoming them would mean increased success of liver transplantation in patients.

Comments on the Quality of English Language

Minor editing needed.

Author Response

Reviewer #3: This is a nice review manuscript on the challenges of liver transplantation for hepatocellular carcinoma, covering many topics, including the use of immunotherapy for immunosuppression. Here are some comments.

  1. It will be nice to include a section on why liver transplantation is one of the primary treatment of hepatocellular carcinoma rather than other therapies such as chemotherapy.
  2. The authors could better explain why challenges they choose to highlight are essential and whether overcoming them would mean increased success of liver transplantation in patients.

R: Thank you very much for your comments and suggestions.

  1. In the introduction (lines 47-49), we emphasize that LT is the therapeutic option with the greatest benefit for overall survival in HCC.
  2. Following your suggestion, we have added a final sentence highlighting the importance of successfully addressing these challenges in this context.

Reviewer 4 Report

Comments and Suggestions for Authors

Dear Authors,

In my opinion, this is very well written article, with use of very nice and understandable English language. The subject is extremely important and it is also one of the most hot topics for HCC and LT specialists ... As a reviewer, I strongly recommend this paper for publication. I sincerely congratulate you on such comprehensive and cross-sectional analysis in your work, especially with a use of the most recent and accurate bibliography :)

I have only two minor comments:

1.     I haven’t found the full-length name for “WL” abbreviation – I guess it’s “Waiting List”, but I think it must be cleared …

2.     Lines 245-246 – In my opinion, it would be nice to have here some examples of names for other locoregional techniques – maybe some of your readers woun’t be specialists in that matter – please, think about it 

Best Regards & once more – Congratulations :)

MW

Author Response

Reviewer #4: In my opinion, this is very well written article, with use of very nice and understandable English language. The subject is extremely important and it is also one of the most hot topics for HCC and LT specialists ... As a reviewer, I strongly recommend this paper for publication. I sincerely congratulate you on such comprehensive and cross-sectional analysis in your work, especially with a use of the most recent and accurate bibliography.

I have only two minor comments:

  1. I haven’t found the full-length name for “WL” abbreviation – I guess it’s “Waiting List”, but I think it must be cleared …
  2. Lines 245-246 – In my opinion, it would be nice to have here some examples of names for other locoregional techniques – maybe some of your readers woun’t be specialists in that matter – please, think about it.

R: Thank you very much for your comments. We have added the explanation for the abbreviation "WL" at its first occurrence and provided examples of other locoregional techniques used for downstaging.